# Efficacy of Guardian Cap Soft-Shell Padding on Head Impact Kinematics in American Football: Pilot Findings

**DOI:** 10.3390/ijerph20216991

**Published:** 2023-10-28

**Authors:** Aaron M. Sinnott, Madison C. Chandler, Charles Van Dyke, David L. Mincberg, Hari Pinapaka, Bradley J. Lauck, Jason P. Mihalik

**Affiliations:** 1Matthew Gfeller Center, Department of Exercise and Sport Science, The University of North Carolina at Chapel Hill, Chapel Hill, NC 27599, USA; asinnott@unc.edu (A.M.S.); hkpinapaka13@gmail.com (H.P.);; 2Campus Health Services, The University of North Carolina at Chapel Hill, Chapel Hill, NC 27599, USA

**Keywords:** brain injury, concussion, intervention strategies, injury prevention, repetitive head impact exposure, equipment add-on

## Abstract

Sport-related concussion prevention strategies in collision sports are a primary interest for sporting organizations and policy makers. After-market soft-shell padding purports to augment the protective capabilities of standard football helmets and to reduce head impact severity. We compared head impact kinematics [peak linear acceleration (PLA) and peak rotational acceleration (PRA)] in athletes wearing Guardian Cap soft-shell padding to teammates without soft-shell padding. Ten Division I college football players were enrolled [soft-shell padding (SHELL) included four defensive linemen and one tight end; non-soft-shell (CONTROL) included two offensive linemen, two defensive linemen, and one tight end]. Participants wore helmets equipped with the Head Impact Telemetry System to quantify PLA (g) and PRA (rad/s^2^) during 14 practices. Two-way ANOVAs were conducted to compare log-transformed PLA and PRA between groups across helmet location and gameplay characteristics. In total, 968 video-confirmed head impacts between SHELL (*n* = 421) and CONTROL (*n* = 547) were analyzed. We observed a Group x Stance interaction for PRA (F_1,963_ = 7.21; *p* = 0.007) indicating greater PRA by SHELL during 2-point stance and lower PRA during 3- or 4-point stances compared to CONTROL. There were no between-group main effects. Protective soft-shell padding did not reduce head impact kinematic outcomes among college football athletes.

## 1. Introduction

The potential short- and long-term risks associated with sustaining sport-related concussion and repetitive head impact exposure (RHIE) are significant concerns for athletes participating in contact and collision sports [1,2]. Public awareness to the implications of contact and collision sport participation has fostered new strategies to reduce the likelihood of concussion onset and the burdens (behavioral, cognitive, and physical) from RHIE in collision sports such as American football [3,4]. As such, understanding how personal protective equipment can effectively reduce head impact severity in football is a primary interest for clinical researchers, healthcare providers, and sport administrators. Soft-shell padding has emerged to potentially reduce head impact severity and concussion risk in a variety of contact and collision sports including rugby [5], soccer [6], and American football [7,8]. The Guardian Cap is a type of soft-shell padding that is affixed to a standard American football helmet and is intended to attenuate forces before being transmitted to the helmet. Considering this, the National Football League mandated that offensive and defensive linemen, linebackers, and tight ends wear the Guardian Cap during the initial weeks of the 2022 preseason. The decision to implement the mandate aligns with evidence suggesting that these position groups experience greater RHIE during practices and games relative to other position groups [9,10], and preliminary evidence suggesting that the preseason may also be a critical period of heightened vulnerability to concussion [11]. That is, greater RHIE in the preseason may increase the likelihood for sport-related concussion diagnosis during the preseason and throughout the remainder of the competitive season [11,12]. While many institutions are implementing policies and mandates relating to the use of soft-shell padding in the interest of preventing injury risk, the efficacy of these devices has not been extensively investigated in field-based settings. The results from these investigations fulfill an imperative step in identifying successful injury-prevention strategies before widespread initiatives are deployed across contact and collision sports.

Although the National Football League reported a 50% reduction in concussion incidence during the 2022 preseason following the mandate for particular position groups to wear the Guardian Cap for team practices [13], most evidence supporting the utility of Guardian Cap was yielded from laboratory-based investigations [14,15]. The Guardian Cap reduced head impact severity during a helmet-drop test at only 2 of 6 helmet locations at high speed (5.47 m/s), and no reductions were observed following low (3.46 m/s) and medium (4.89 m/s) drop speeds across all helmet locations [15]. An adaptation by Bailey et al. examined head impact severity from two helmets colliding at various angles replicative of gameplay with various helmet model and soft-shell padding combinations [14]. They observed an average reduction in head impact severity by 9% compared to helmets not equipped with the Guardian Cap, and upwards of 20% when both helmets had a Guardian Cap [14]. From these early findings, it is important for researchers to consider both the helmet’s location of impact in addition to potential factors in gameplay scenarios that may influence the efficacy of soft-shell padding.

To date, there is limited prospective evidence from field-based studies examining the impact of soft-shell padding on head impact kinematics. Peak linear acceleration (PLA) and peak rotational acceleration (PRA) are common kinematic outcome measures employed in studies to quantify head impact severity and have been used in studies addressing soft-shell padding. Quigley et al. recently observed no significant differences in PLA or PRA among seven players who participated in three practices without a Guardian Cap compared to three practices with the Guardian Cap [8]. Similarly, five linebackers with traditional helmets across 13 practices in 2019 sustained head impacts of equivocal severity when compared to five linebackers using the Guardian Cap across 14 practices during the 2021 season [7]. Although both studies were able to appropriately video-confirm head impacts, video review can also open the aperture to examining the influence of other gameplay characteristics that might influence head impact kinematics. These include the observed direction of impact initiation (front, left, right), starting stance (2-, 3-, or 4-point stance), closing distance (less than or greater than 5 yards), skill (blocking, tackling, etc.), striking (initiating impact vs. receiving impact), and play-type [16,17,18,19]. Overall, evidence examining head impact severity with the Guardian Cap in a field-based setting is sparse, and there is a need to examine gameplay characteristics unique to American football to further inform the efficacy of soft-shell padding to reduce head impact severity and overall injury risk.

Effective primary and tertiary injury-prevention strategies can improve safety in contact and collision sports but there is sparse evidence examining secondary prevention approaches to reduce sport-related concussion (SRC) and RHIE. Studying soft-shell padding to effectively reduce head impact kinematics in a field-based study is an important advance for researchers and healthcare providers interested in improving the health and safety of college athletes. A socioeconomic framework has previously been used in concussion literature to contextualize the intra- and inter-personal factors contributing to injury onset, but effective initiatives to prevent SRC in collision sports remain a scientific burden for public health officials. We aimed to compare PLA and PRA between college football players with soft-shell padding to teammates from similar positions who chose not to wear soft-shell padding during preseason practices. Further, we aimed to examine the helmet location of sustained impact (e.g., front, side, top, or rear of helmet) in addition to gameplay characteristics to elucidate if contextual factors from football participation affect the efficacy of soft-shell padding. Like preceding reports [7,8], we hypothesized that soft-shell padding would not reduce head impact severity.

## 2. Materials and Methods

### 2.1. Research Design and Participants

We conducted a prospective observational study during the 2022 competitive season and examined 14 full-contact practices preceding the first competition. In total, ten Division I college football players either wore soft-shell padding [SHELL; *n* = 5; 4 defensive linemen and 1 tight end; height = 195.4 ± 2.2 cm; mass = 125.6 ± 13.3 kg] or chose not to wear soft-shell padding [CONTROL; *n* = 5; 2 offensive linemen, 2 defensive linemen, and 1 tight end; height = 189.5 ± 5.5 cm; mass = 120.2 ± 8.3 kg].

### 2.2. Instrumentation

The Guardian Cap NXT (Guardian Innovations; Corners, GA, USA) consists of a dual-layer foam pad that encapsulates the exterior of a standard American football helmet. The padding can be attached or removed with the use of buttons to secure the padding to the facemask, and a Velcro pad to ensure a secure fit with the helmet (Figure 1).

### 2.3. Measurements/Outcomes

Standard issue Riddell Speed Flex helmets (Riddell Sports Group, Rosemont, IL, USA) were equipped with the Head Impact Telemetry (HIT) System (Riddell, Elyria, OH, USA), which consists of an array of six spring-loaded single-axis accelerometers and the Sideline Response System (Figure 2).

Once a single accelerometer detected a linear acceleration exceeding 9.6 g, data from all 6 accelerometers sampled linear and rotational accelerations at 1000 Hz for 40 ms (8 ms pre-trigger and 32 ms post-trigger) to compute resultant PLA and PRA (measured in g’s and rad/s^2^, respectively) and impact location as primary outcome measures. The measurement error is approximately 15% for linear and rotational acceleration [20]. These data were then date- and time-stamped and transmitted to the Sideline Response System. In special circumstances when the real-time data transmission was absent (e.g., signal interruptions, Sideline Response System not set up, etc.), head impacts were locally stored in non-volatile memory built into the monitoring system and transmitted to the Sideline Response System later.

At each practice, 2 video cameras (Canon Vixia HG10 AVCHD Camcorder, Canon, Tokyo, Japan) were positioned on an elevated platform [20] ft (6.1 m) and recorded video (24 frames per second) of offensive and defensive linemen position groups in addition to team-based activities throughout practice. A universal timeclock was presented at the outset of each video to synchronize recorded head impacts with video. Similar to prior analyses of video-confirmed head impacts and gameplay characteristics in youth ice hockey [21], and high school football (Sinnott et al., in review), one research study team member (C.V.) with extensive film review experience video-verified true positive head impacts. False-negative impacts were not reviewed. All relevant data from video-confirmed head impacts were extracted with the use of an online survey platform (Qualtrics) and further categorized based on the observed direction, starting stance, closing distance, striking, Guardian Cap impacts (i.e., whether one, both, or neither player was wearing a Guardian Cap during the play), and play-type for final analysis (Table 1).

### 2.4. Procedures

All participants provided their informed and voluntary consent in accordance with the Declaration of Helsinki, and the protocol was approved by The University of North Carolina’s Institutional Review Board (Protocol No: 21-1960). Soft-shell padding was affixed to helmets in the SHELL group prior to the first practice. Soft-shell padding and HIT sensor usage was recorded daily, and correct helmet placement was evaluated weekly.

### 2.5. Data Processing and Statistical Analysis

Participants completed a full season with HIT System data, and impacts recorded outside practice sessions were removed from the dataset. We also excluded events sustained during helmets-only practices, impacts from special teams, screen plays, bag drills, shoulder-pad-initiated impacts, and statistical outliers (≥3 standard deviations from group median) with respect to each participant. Due to the non-normality of PLA and PRA outcomes because head impacts skew to lower magnitudes, median and interquartile ranges were calculated, and these outcomes underwent natural logarithmic transformations. Separate two-way ANOVAs compared PLA and PRA between CONTROL and SHELL groups across helmet location and gameplay characteristics. Post-hoc analyses underwent Bonferroni corrections, and all analyses were conducted with SPSS (IBM, Version 28) with an a priori alpha level of 0.05. The computer code used for analyses is available from the authorship team upon reasonable request.

## 3. Results

In total, 2653 impacts were recorded and underwent video review. Of 1225 video-confirmed impacts, we excluded 192 (15.67%) impacts from helmets-only sessions, 42 (3.43%) from special teams, screen plays, and bag drills, 14 (1.14%) shoulder-pad-initiated impacts, and 9 (0.73%) statistical outliers (no more than 2 impacts per player). A final sample of 968 impacts underwent final analyses. Chi-square tests revealed equal distributions between SHELL and CONTROL groups across helmet location, observed direction, starting stance, and closing distance (*p* values > 0.05; Table 2).

However, likely as a result of unequal position-matching, the impacts between SHELL and CONTROL groups were unequally distributed across skill, striking, Guardian Cap impacts, and play-type factors (all *p* values < 0.001). Due to the small sample size and preliminary nature of the current study, these factors were not included in the final analyses. See Table 3 for descriptive statistics (split by CONTROL and SHELL groups) of PLA and PRA values across all independent variable conditions.

Overall, PLA and PRA between SHELL and CONTROL were equivocal across helmet location, observed direction, starting stance, and closing distance gameplay factors (*p* > 0.05).

We observed a main effect for helmet location across both PLA (*p* < 0.001, η*_p_*^2^ = 0.041), and PRA (*p* = 0.008, η*_p_*^2^ = 0.082) outcomes. Post hoc analyses revealed a greater PLA to the front of the helmet compared to the left (*p* < 0.001) and right (*p* = 0.002) sides; and top/crown impacts resulted in a greater PLA than impacts to the left (*p* < 0.001), right (*p* = 0.002), and back (*p* = 0.010) of the helmet. The PRA from impacts sustained to the helmet’s front was also greater than impacts to the left (*p* = 0.04), right (*p* = 0.009), and top (*p* < 0.001) of the helmet. Similarly, there was a main effect of observed direction for both PLA (*p* = 0.004, η*_p_*^2^ = 0.011) and PRA (*p* = 0.029, η*_p_*^2^ = 0.007). Post hoc analyses revealed a greater PLA from impacts sustained to the front of the helmet than the left (*p* = 0.028) or right (*p* = 0.024) side impacts, and a greater PRA than right (*p* = 0.04) side impacts.

Among gameplay characteristics, we observed a Group x Stance interaction for PRA (*p* = 0.007, η*_p_*^2^ = 0.007) indicating a greater PRA by SHELL during the 2-point stance and lower PRA during the 3- or 4-point stances compared to CONTROL. There were also main effects for the starting stance for both PLA (*p* = 0.006, η*_p_*^2^ = 0.008) and PRA (*p* < 0.001, η*_p_*^2^ = 0.034) with the 2-point stance resulting in greater acceleration than the 3- or 4-point stance. Lastly, main effects of closing distance were also observed whereby distances greater than 5 yards resulted in a greater PLA (*p* < 0.001, η*_p_*^2^ = 0.021) and PRA (*p* = 0.001, η*_p_*^2^ = 0.016) than closing distances shorter than 5 yards. Statistical results of ANOVA analyses are reported in Table 4.

## 4. Discussion

Protective equipment design and modification are key areas of interest for environmental health researchers seeking to promote cost-effective and evidence-based injury prevention strategies in collision sports. Based on our preliminary findings of head impact exposure during the Fall 2022 season, Guardian Caps did not affect head impact kinematic outcomes among offensive and defensive linemen and tight ends. Our findings are supported by Quigley et al. who observed no effect of the Guardian Cap in PLA and PRA among seven players with complete data from six practice sessions [8]. More specifically, one offensive linemen, one defensive linemen, one running back, one tight end, and three linebackers completed team-based practice drills for an initial three practices (without Guardian Cap) and three follow-up sessions (with Guardian Cap), no changes were observed in PLA or PRA. Notably, our findings with 10 offensive and defensive linemen and tight ends examined the helmet location of sustained impacts in addition to gameplay characteristics that contribute to head impact kinematics in college football.

In addition to the 2-point stance resulting in a greater PLA and PRA than the 3-point stance, we observed a greater PRA with soft-shell padding during the 2-point stance and a lower PRA during the 3- or 4-point stances when compared to traditional helmets. This finding contradicts a previous report that linear and rotational accelerations were similar between the 2-point and 3-point stances [22], in addition to a study that found a greater head impact exposure with the 3-point stance compared to the 2-point stance [23]. Our findings may be in part due to positional differences between SHELL and CONTROL groups. In the current study, four (of five) players that chose to wear soft-shell padding were defensive linemen and no offensive linemen elected to wear soft-shell padding, whereas only two chose to wear soft-shell padding in the CONTROL group. We also observed main effects for helmet location, observed direction, and closing distance, which builds upon previous studies indicating that these factors contribute to our understanding of RHIE in college football [17,19,21,22].

Although the proportion of impacts between SHELL and CONTROL groups were unequally distributed across skill, striking, Guardian Cap impacts, and play-type factors, their inclusion in the current study is important as part of a preliminary investigation of head impact kinematics with soft-shell padding in a field-based setting. To date, there are a limited number of field-based studies examining head impact severity with the use of after-market soft-shell padding, but the current report is the first examining gameplay factors between closely-matched collegiate football athletes throughout preseason practices. As such, the collective author group advocates further investigations to account for these contextual factors that may affect head impact kinematics. Our findings also coincide with prior studies that observed gameplay factors such as observed direction and closing distance which can influence head impact kinematics [17,18,19,23]. Although the current study investigated head impact kinematics and not concussion incidence, our findings provide additional information to consider in light of a recent report that concussions among professional football athletes were reduced by upwards of 50% with the use of soft-shell padding [13]. Notably, those findings were not peer-reviewed, and it is unknown if athlete exposures and other contextual factors were controlled in their analyses.

### Implications for Injury Prevention

There are several notable implications for our findings in larger public health that can extend from prior laboratory-based studies and be addressed with future research. Most concussions in college football occur in practice [4], which may be a viable setting to implement injury prevention strategies for concussion and overall RHIE. Our study involved the Riddell Speed Flex (Riddell Sports Group, Rosemont, IL, USA) football helmets, which have been used in prior field-based studies of the Guardian Cap but have not been previously used in several laboratory-based studies [8]. Helmet speed preceding impacts can also influence the efficacy of soft-shell padding [15], and we were unable to determine the speed of helmets worn by players. Incorporating GPS-tracking instrumentation can quantify player speed preceding head impacts and facilitate the translation between lab- and field-based studies of head impact kinematics. The associated costs to deploy soft-shell padding as equipment add-ons should also be considered before widespread recommendation and deployment to prevent concussion occurrence. Future work with larger sample sizes and from more than one study site may be able to inform alternative analytical approaches (intention to treat analyses, etc.) for sport administrators considering the implementation of soft-shell padding within a sporting body. Information from these investigations can further evaluate the football specific contexts in which soft-shell protective padding may best reduce the head impact burden for athletes who choose to use them.

Our findings build upon previous recommendations for sports injury research to examine contextual factors that may contribute to injury risk within the socioecological model [24,25]. Future research examining athlete perceptions in addition to interpersonal, organizational, environmental, and policy aspects of the sociological framework will elucidate underlying factors that may contribute to successful injury prevention in collision sports. Overall, our null findings should also be considered with recent recommendations set forth by the National Athletic Trainers’ Association position statement to reduce head-first contact behavior in American football. Particularly, (in item #11) to recognize that after-market companies may overstate injury prevention benefits, which may lead to altered risk-taking behavior [26]. Although we were unable to account for potential behavioral differences for participants choosing to wear a Guardian Cap compared to those without soft-shell padding, risk-taking behavior among athletes under the pretense of increased protection should be considered in future studies examining soft-shell padding. Future studies examining individuals across practices and games (with soft-shell padding and without, respectively) may be able to elucidate if behavioral differences between settings influence the efficacy of soft-shell padding. The author team acknowledges that injury-mitigation strategies for medical professionals are multifaceted and consider contextual factors (competitive level, history of prior concussions, etc.) other than the assigned position group to make recommendations for protective equipment. And particularly for lower competitive levels such as high school or youth settings with fewer available resources, the type and age of standard football helmets are also important for clinicians interested in augmenting the protective capabilities of standard equipment. These preliminary results also coincide with recent Consensus Head Acceleration Measurement Practices (CHAMP) recommendations to improve the scientific rigor and quality of investigations for head impact exposure in sport (Appendix A) [27]. The current study is one of the first to disclose key study characteristics in accordance with CHAMP recommendations and will in turn inform interpretation and replication of future investigations examining head impact biomechanics in sport.

Several limitations should be considered with our study findings to be generalized across all football populations including youth, college, and professional levels. Our results are limited to a small sample of college football players wearing one type of football helmet in one program. Future studies among various age ranges utilizing additional protective equipment such as instrumented mouthpieces may be able to differentiate Guardian Cap efficacy across various helmet makes and models. We were also unable to examine linebackers, a position group included in the NFL mandate and included in prior studies [7,8]. It is unknown if our findings are generalizable to this position group because of differing positional roles and overall RHIE [17]. Although multiple cameras were deployed to capture most head impacts, there is the possibility that some head impacts were sustained outside the field of view of the cameras or were not confirmed during video review and thus omitted in the current study. Lastly, the soft-shell padding is available in several contact and collision sports aside from American football and the current study is limited to only one sport. Addressing these shortcomings in future work can improve our understanding of efficacy of after-market soft-shell padding in various sports and competitive levels.

## 5. Conclusions

Based on our preliminary findings, protective soft-shell padding did not reduce head linear or rotational accelerations among a subset of college football players. Additionally, gameplay characteristics fundamental to offensive linemen, defensive linemen, and tight ends did not affect peak linear and rotational acceleration between those with and without soft-shell padding. Intra and interpersonal contextual factors that contribute to RHIE and concussion occurrence in collision sports should be considered in future studies evaluating the efficacy of soft-shell padding.

## Figures and Tables

**Figure 1 ijerph-20-06991-f001:**
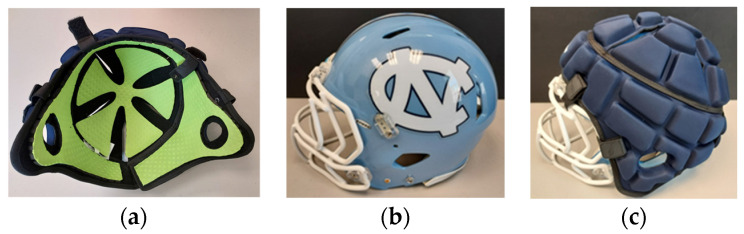
Photographs depicting the (**a**) inner lining of Guardian Cap soft-shell padding, (**b**) CONTROL helmet condition, and (**c**) SHELL helmet condition.

**Figure 2 ijerph-20-06991-f002:**
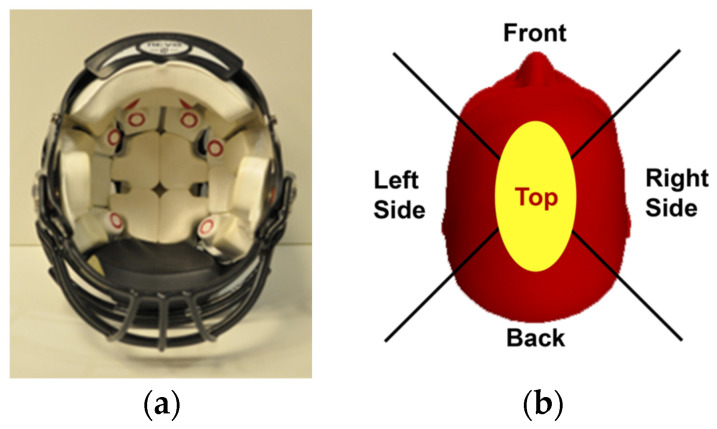
Photographs depicting (**a**) sensor array embedded inside a standard Riddell Speed Flex football helmet, and (**b**) orthogonal lines to categorize impact location.

**Table 1 ijerph-20-06991-t001:** Gameplay characteristics categorized from video-confirmed head impacts.

Variable	Category
Helmet Location	Front
	Left
	Right
	Top/Crown
	Back
Observed Direction	Front
	Left
	Right
Starting Stance	2-point
	3- or 4-point
Closing Distance	≤5 yards
	>5 yards
Skill	Blocking
	Engaging
	Tackling
Striking	Striking Player
	Player Struck
Guardian Cap Impacts	None
	Yes, 1 Player
	Yes, Both Players
Play-type	Pass
	Rush
	Individual
	Inside Run
	1 vs. 1

Table 1 Categorical variables obtained from helmet-embedded accelerometers and video review.

**Table 2 ijerph-20-06991-t002:** Frequency and percentage (%) of sustained head impacts and results of chi-square analyses between the CONTROL and SHELL groups.

Variable	Category	CONTROL (*n* = 547)	SHELL (*n* = 421)	χ^2^ (df), *p* Value
Helmet Location	Front	446 (81.5%)	340 (80.8%)	6.95 (4), 0.139
	Left	29 (5.3%)	21 (5.0%)	
	Right	41 (7.5%)	29 (6.9%)	
	Top/Crown	7 (1.3%)	16 (3.8%)	
	Back	24 (4.4%)	15 (3.6%)	
Observed Direction	Front	458 (84.0%)	359 (86.1%)	1.01 (2), 0.605
	Left	38 (6.9%)	61 (5.5%)	
	Right	49 (8.9%)	35 (8.4%)	
Starting Stance	2-point	82 (15.0%)	76 (18.1%)	1.60 (1), 0.206
	3-point	464 (85.0%)	345 (81.9%)	
Closing Distance	≤5 yards	526 (96.2%)	397 (94.3%)	0.86 (1), 0.173
	>5 yards	21 (3.8%)	24 (5.7%)	
Skill	Blocking	327 (59.8%)	65 (15.4%)	215.30 (2), <0.001
	Engaging	197 (36.0%)	350 (83.1%)	
	Tackling	23 (4.2%)	6 (1.5%)	
Striking	Strike	210 (38.4%)	261 (61.9%)	53.06 (1), <0.001
	Struck	337 (61.6%)	160 (38.0%)	
Guardian Cap Impacts	None	493 (90.1%)	-	770.35 (2), <0.001
	Yes, 1 Player	54 (9.9%)	400 (95.0%)	
	Yes, Both Players	-	21 (5.0%)	
Play-type	Pass	81 (14.8%)	96 (22.8%)	226.30 (4), <0.001
	Rush	103 (18.8%)	152 (36.1%)	
	Individual	262 (47.9%)	18 (4.2%)	
	Inside Run	77 (14.1%)	99 (23.5%)	
	1 vs. 1	24 (4.4%)	56 (13.3%)	

Table 2 Distribution of head impacts between groups were comparable for the measured location in addition to observed direction, starting stance, and closing distance categories.

**Table 3 ijerph-20-06991-t003:** Median [interquartile range] of peak linear acceleration (PLA; g) and peak rotational acceleration (PRA; rad/s^2^) between CONTROL and SHELL groups across location of measured head Impacts and gameplay characteristics.

Variable	Factor	CONTROL (*n* = 547)		SHELL (*n* = 421)	
		PLA	PRA	PLA	PRA
Helmet Location	Front	25.20 [17.75]	1805.10 [1258.54]	24.55 [15.47]	1605.63 [1007.14]
	Left	15.60 [14.25]	1291.93 [764.70]	18.30 [10.15]	1434.47 [864.92]
	Right	19.30 [11.30]	1485.84 [775.80]	19.00 [13.15]	1395.71 [856.74]
	Top/Crown	38.20 [34.45]	883.77 [1619.75]	28.31 [22.47]	896.65 [887.62]
	Back	19.85 [9.95]	1254.27 [692.97]	22.70 [10.75]	999.38 [758.73]
Observed Direction	Front	24.80 [17.21]	1782.04 [1234.63]	24.11 [15.53]	1557.53 [995.92]
	Left	17.25 [14.62]	1383.78 [885.91]	24.80 [13.31]	1520.97 [1142.53]
	Right	22.32 [12.35]	1485.84 [938.21]	20.61 [14.21]	1440.92 [1043.86]
Starting Stance	2-point	25.35 [24.45]	1829.49 [1820.53]	27.90 [23.52]	2189.93 [1752.81]
	3-point	23.70 [16.85]	1642.07 [1147.51]	23.44 [14.55]	1487.22 [888.95]
Closing Distance	≤5 yards	23.40 [16.93]	1650.96 [1154.83]	23.73 [15.15]	1529.66 [966.23]
	>5 yards	36.62 [26.25]	2889.21 [2351.35]	30.15 [32.87]	2336.86 [1530.25]
Skill	Blocking	22.65 [16.75]	1677.50 [1332.35]	29.21 [22.03]	2251.74 [1796.01]
	Engaging	25.31 [18.42]	1695.63 [1062.80]	23.45 [15.02]	1487.10 [891.26]
	Tackling	21.50 [21.02]	1493.91 [1442.02]	20.92 [12.52]	1400.58 [1528.51]
Striking	Strike	26.71 [21.05]	1821.63 [1322.26]	23.91 [16.15]	1518.54 [971.27]
	Struck	22.35 [15.12]	1569.63 [1168.09]	23.80 [13.77]	1595.56 [1109.25]
Guardian Cap Impacts	None	24.60 [17.27]	1712.83 [1247.18]	-	-
	Yes, 1 Player	22.30 [10.81]	1699.71 [1250.72]	23.90 [15.50]	1535.03 [1027.92]
	Yes, both Players	-	-	25.85 [14.30]	1773.37 [944.34]
Play-type	Pass	21.50 [14.00]	1562.38 [1021.15]	22.10 [11.71]	1430.71 [831.45]
	Rush	28.60 [19.62]	1950.49 [1427.95]	25.35 [17.43]	1628.56 [1145.83]
	Individual	22.10 [15.35]	1625.98 [1149.98]	20.51 [21.22]	1458.71 [1460.31]
	Inside Run	25.90 [18.95]	1763.53 [1556.58]	24.90 [15.02]	1748.84 [1295.24]
	1 vs. 1	26.91 [19.17]	1772.92 [1110.48]	23.42 [15.37]	1393.53 [653.31]

Abbreviations: PLA, peak linear acceleration; PRA, peak rotational acceleration.

**Table 4 ijerph-20-06991-t004:** Resultant models from two-way ANOVA comparing CONTROL and SHELL groups across peak linear acceleration (PLA) and peak rotational acceleration (PRA) outcomes across impact location and gameplay characteristics.

Factor	ANOVA Model	Outcomes			
		Linear Acceleration		Rotational Acceleration	
		F Ratios and *p* Values	Effect Size (η*_p_*^2^)	F Ratios and *p* Values	Effect Size (η*_p_*^2^)
Helmet Location	GROUP (CONTROL vs. SHELL)	F_1,958_ = 0.18; *p* = 0.758	0.001	F_1,958_ = 0.90; *p* = 0.342	0.001
	Location-Measured	F_4,958_ = 10.35; *p* < 0.001	0.041	F_4,958_ = 17.44, *p* = 0.008	0.082
	GROUP X Location-Measured	F_4,958_ = 0.18; *p* = 0.947	0.001	F_4,958_ = 1.23, *p* = 0.297	0.005
Observed Direction	GROUP (CONTROL vs. SHELL)	F_1,956_ = 0.92; *p* = 0.337	0.001	F_1,956_ = 0.11; *p* = 0.740	0.001
	Direction-Observed	F_2,956_ = 5.59; *p* = 0.004	0.011	F_2,956_ = 3.59; *p* = 0.029	0.007
	GROUP X Direction-Observed	F_2,956_ = 1.50; *p* = 0.223	0.003	F_2,956_ = 0.98; *p* = 0.375	0.002
Starting Stance	GROUP (CONTROL vs. SHELL)	F_1,963_ = 1.58; *p* = 0.209	0.966	F_1,963_ = 0.02; *p* = 0.724	0.001
	Stance	F_1,963_ = 7.68; *p* = 0.006	0.008	F_1,963_ = 4.70; *p* < 0.001	0.034
	GROUP X Stance	F_1,963_ = 2.52; *p* = 0.113	0.003	F_1,963_ = 7.21; *p* = 0.007	0.007
Closing Distance	GROUP (CONTROL vs. SHELL)	F_1,964_ = 0.15; *p* = 0.698	0.001	F_1,964_ = 5.58; *p* = 0.209	0.002
	Closing Distance	F_1,964_ = 20.33; *p* < 0.001	0.021	F_1,964_ = 15.40; *p* < 0.001	0.016
	GROUP X Closing Distance	F_1,964_ = 0.23; *p* = 0.592	0.001	F_1,964_ = 0.01; *p* = 0.918	0.001

Abbreviations: PLA, peak linear acceleration; PRA, peak rotational acceleration. Table 4 Peak linear and rotational acceleration were comparable between players wearing after-market soft-shell padding to those without padding.

## Data Availability

The data presented in this study may be available on request from the corresponding author. The data are not publicly available given the small sample size and the potential to directly or indirectly identify individual study participants.

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
