# Peer review of "Efficacy of Guardian Cap Soft-Shell Padding on Head Impact Kinematics in American Football: Pilot Findings"

_ijerph, 2023, doi:10.3390/ijerph20216991_

Round 1

Reviewer 1 Report

Comments and Suggestions for Authors

Thank you very much for the opportunity to review such an interesting manuscript. The authors address a topic that has been developed over the last few years in a critical and systematic manner.

Introduction

It provides the relevant and necessary information for the understanding of the research area covered by the scientific article.

Material and Methods

The information is correct and the methodology of the study is clear, although I would recommend the authors to create sub-sections, for example "2.1. Research Design and Participants" etc.

Results

I recommend using the journal format for titles and table captions. 

Discussion and Conclusions

Correct and adequate

Reviewer 2 Report

Comments and Suggestions for Authors

The following items are suggested for further revision.

1.      Line 91, First time to see “SRC” which wasn’t mentioned before without clearly stating the full term

2.      Line 120, Figures of helmets and accelerometers’ positions will be better for the reader's understanding of the definition of impact location (i.e. Front, left, right, top, back)

3.      Line 179, the median of each variable was selected for further statistical analysis without any explanation or literature support.

Reviewer 3 Report

Comments and Suggestions for Authors

Thank you for the opportunity to review your manuscript

Author Response

Thank you for the opportunity to review your manuscript.

            The author team would like to sincerely thank you for considering our paper a benefit to the greater public health and scientific communities.

Reviewer 4 Report

Comments and Suggestions for Authors

This is a well written paper and reads smoothly. Prior to publication I would like the author to make some minor edits. 

1) Include the international units beside the American units, such as 20ft for the height of the camera should also be in meters. 

2) Provide figures with the CONTROL and SHELL conditions, in addition to a brief figure showing how the HIT system is embedded. 

3) Add some practical applications for clinicians on how the information in this study could help 

4) Is there any laboratory based studies testing similar conditions? 

5) Explain the reasoning why you didn't calculate HIC15ms or HIP or other head injury criterion. 

Reviewer 5 Report

Comments and Suggestions for Authors

Good material, well-presented.

I have marked 'Interest to readers' as 'Average', only because that material is quite specific to a single sports discipline. And you did not even mention that your findings could be of interest/value to other sports, where protection helmets are involved.

Author Response

   The author team would like to sincerely thank you for considering our paper a benefit to the greater public health and scientific communities.